SCIENCE FORUM

# Wikidata as a knowledge graph for the life sciences

**Abstract** Wikidata is a community-maintained knowledge base that has been assembled from repositories in the fields of genomics, proteomics, genetic variants, pathways, chemical compounds, and diseases, and that adheres to the FAIR principles of findability, accessibility, interoperability and reusability. Here we describe the breadth and depth of the biomedical knowledge contained within Wikidata, and discuss the open-source tools we have built to add information to Wikidata and to synchronize it with source databases. We also demonstrate several use cases for Wikidata, including the crowdsourced curation of biomedical ontologies, phenotype-based diagnosis of disease, and drug repurposing.

ANDRA WAAGMEESTER[†], GREGORY STUPP[†], SEBASTIAN BURGSTALLER-MUEHLBACHER, BENJAMIN M GOOD, MALACHI GRIFFITH, OBI L GRIFFITH, KRISTINA HANSPERS, HENNING HERMJAKOB, TOBY S HUDSON, KEVIN HYBISKE, SARAH M KEATING, MAGNUS MANSKE, MICHAEL MAYERS, DANIEL MIETCHEN, ELVIRA MITRAKA, ALEXANDER R PICO, TIMOTHY PUTMAN, ANDERS RIUTTA, NURIA QUERALT-ROSINACH, LYNN M SCHRIML, THOMAS SHAFEE, DENISE SLENTER, RALF STEPHAN, KATHERINE THORNTON, GINGER TSUENG, ROGER TU, SABAH UL-HASAN, EGON WILLIGHAGEN, CHUNLEI WU AND ANDREW I SU*

## Introduction

Integrating data and knowledge is a formidable challenge in biomedical research. Although new scientific findings are being discovered at a rapid pace, a large proportion of that knowledge is either locked in data silos (where integration is hindered by differing nomenclature, data models, and licensing terms; *Wilkinson et al., 2016*) or locked away in free-text. The lack of an integrated and structured version of biomedical knowledge hinders efficient querying or mining of that information, thus preventing the full utilization of our accumulated scientific knowledge.

Recently, there has been a growing emphasis within the scientific community to ensure all scientific data are FAIR – Findable, Accessible, Interoperable, and Reusable – and there is a growing consensus around a concrete set of principles to ensure FAIRness (*Wilkinson et al., 2019*; *Wilkinson et al., 2016*). Widespread implementation of these principles would greatly

advance efforts by the open-data community to build a rich and heterogeneous network of scientific knowledge. That knowledge network could, in turn, be the foundation for many computational tools, applications and analyses.

Most data- and knowledge-integration initiatives fall on either end of a spectrum. At one end, centralized efforts seek to bring multiple knowledge sources into a single database (see, for example, *Mungall et al., 2017*): this approach has the advantage of data alignment according to a common data model and of enabling high performance queries. However, centralized resources are difficult and expensive to maintain and expand (*Chandras et al., 2009*; *Gabella et al., 2018*), at least in part because of bottlenecks that are inherent in a centralized design.

At the other end of the spectrum, distributed approaches to data integration result in a broad landscape of individual resources, focusing on technical infrastructure to query and integrate

*For correspondence: asu@scripps.edu

†These authors contributed equally to this work

**Competing interests:** The authors declare that no competing interests exist.

across them for each query. These approaches lower the barriers to adding new data by enabling anyone to publish data by following community standards. However, performance is often an issue when each query must be sent to many individual databases, and the performance of the system as a whole is highly dependent on the stability and performance of each individual component. In addition, data integration requires harmonizing the differences in the data models and data formats between resources, a process that can often require significant skill and effort. Moreover, harmonizing differences in data licensing can sometimes be impossible.

Here we explore the use of Wikidata (www. wikidata.org; *Vrandečić, 2012*; *Mora-Cantallops et al., 2019*) as a platform for knowledge integration in the life sciences. Wikidata is an openly-accessible knowledge base that is editable by anyone. Like its sister project Wikipedia, the scope of Wikidata is nearly boundless, with items on topics as diverse as books, actors, historical events, and galaxies. Unlike Wikipedia, Wikidata focuses on representing knowledge in a structured format instead of primarily free text. As of September 2019, Wikidata's knowledge graph included over 750 million statements on 61 million items (tools.wmflabs.org/ wikidata-todo/stats.php). Wikidata was also the first project run by the Wikimedia Foundation (which also runs Wikipedia) to have surpassed one billion edits, achieved by a community of 12,000 active users, including 100 active computational 'bots' (*Figure 1—figure supplement 1*).

As a knowledge integration platform, Wikidata combines several of the key strengths of the centralized and distributed approaches. A large portion of the Wikidata knowledge graph is based on the automated imports of large structured databases via Wikidata bots, thereby breaking down the walls of existing data silos. Since Wikidata is also based on a community-editing model, it harnesses the distributed efforts of a worldwide community of contributors, including both domain experts and bot developers. Anyone is empowered to add new statements, ranging from individual facts to large-scale data imports. Finally, all knowledge in Wikidata is queryable through a SPARQL query interface (query.wikidata.org/), which also enables distributed queries across other Linked Data resources.

In previous work, we seeded Wikidata with content from public and authoritative sources of structured knowledge on genes and proteins (*Burgstaller-Muehlbacher et al., 2016*) and chemical compounds (*Willighagen et al., 2018*). Here, we describe progress on expanding and enriching the biomedical knowledge graph within Wikidata, both by our team and by others in the community (*Turki et al., 2019*). We also describe several representative biomedical use cases on how Wikidata can enable new analyses and improve the efficiency of research. Finally, we discuss how researchers can contribute to this effort to build a continuously-updated and community-maintained knowledge graph that epitomizes the FAIR principles.

## The Wikidata Biomedical Knowledge Graph

The original effort behind this work focused on creating and annotating Wikidata items for human and mouse genes and proteins (*Burgstaller-Muehlbacher et al., 2016*), and was subsequently expanded to include microbial reference genomes from NCBI RefSeq (*Putman et al., 2017*). Since then, the Wikidata community (including our team) has significantly expanded the depth and breadth of biological information within Wikidata, resulting in a rich, heterogeneous knowledge graph (*Figure 1*). Some of the key new data types and resources are described below.

**Genes and proteins**: Wikidata contains items for over 1.1 million genes and 940 thousand proteins from 201 unique taxa. Annotation data on genes and proteins come from several key databases including NCBI Gene (*Agarwala et al., 2018*), Ensembl (*Zerbino et al., 2018*), UniProt (*UniProt Consortium, 2019*), InterPro (*Mitchell et al., 2019*), and the Protein Data Bank (*Burley et al., 2019*). These annotations include information on protein families, gene functions, protein domains, genomic location, and orthologs, as well as links to related compounds, diseases, and variants.

**Genetic variants**: Annotations on genetic variants are primarily drawn from CIViC (http://www. civicdb.org), an open and community-curated database of cancer variants (*Griffith et al., 2017*). Variants are annotated with their relevance to disease predisposition, diagnosis, prognosis, and drug efficacy. Wikidata currently contains 1502 items corresponding to human genetic variants,

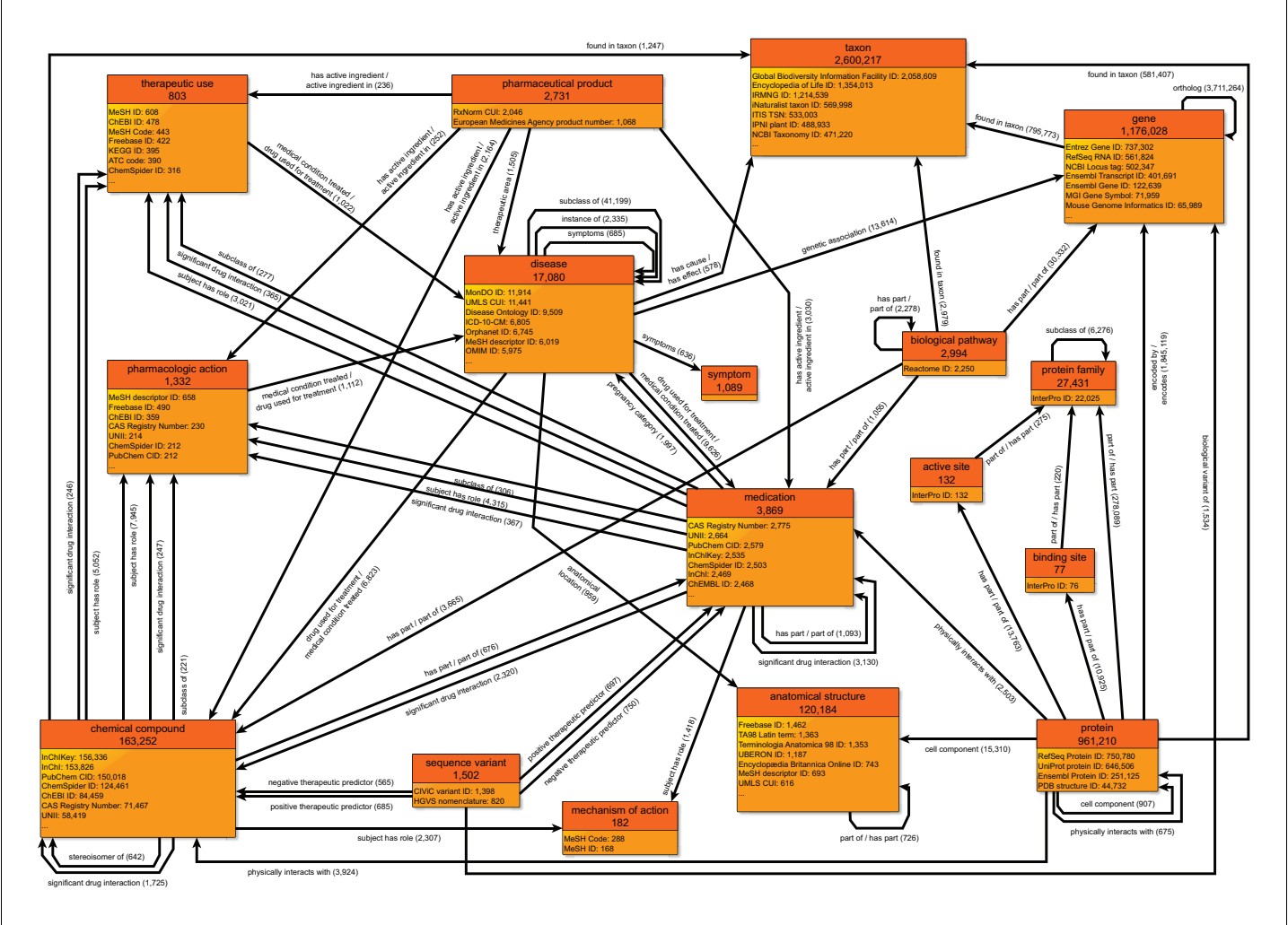

**Figure 1.** A simplified class-level diagram of the Wikidata knowledge graph for biomedical entities. Each box represents one type of biomedical entity. The header displays the name of that entity type (e.g., pharmaceutical product) and the number of Wikidata items for that entity type. The lower portion of each box displays a partial listing of attributes about each entity type and the number of Wikidata items for each attribute. Edges between boxes represent the number of Wikidata statements corresponding to each combination of subject type, predicate, and object type. For example, there are 1505 statements with 'pharmaceutical product' as the subject type, 'therapeutic area' as the predicate, and 'disease' as the object type. For clarity, edges for reciprocal relationships (e.g., 'has part' and 'part of') are combined into a single edge, and scientific articles (which are widely cited in statement references) have been omitted. All counts of Wikidata items are current as of September 2019. The most common data sources cited as references are available in *Figure 1—source data 1*. Data are generated using the code in https://github.com/SuLab/genewikiworld (archived at *Mayers et al., 2020*). A more complete version of this graph diagram can be found at https://commons.wikimedia.org/wiki/File:Biomedical_Knowledge_Graph_in_Wikidata.svg.

The online version of this article includes the following source data and figure supplement(s) for figure 1:

**Source data 1.** Most frequent data sources cited as references for the biomedical subset of the Wikidata knowledge graph shown in *Figure 1*.
**Figure supplement 1.** Trends in Wikidata edits.

focused on those with a clear clinical or therapeutic relevance.

**Chemical compounds including drugs**: Wikidata has items for over 150 thousand chemical compounds, including over 3500 items which are specifically designated as medications. Compound attributes are drawn from a diverse set of databases, including PubChem

(*Wang et al., 2009*), RxNorm (*Nelson et al., 2011*), the IUPHAR Guide to Pharmacology (*Harding et al., 2018*; *Pawson et al., 2014*; *Southan et al., 2016*), NDF-RT (National Drug File – Reference Terminology), and LIPID MAPS (*Sud et al., 2007*). These items typically contain statements describing chemical structure and key physicochemical properties, and links to

databases with experimental data, such as Mass-Bank (*Horai et al., 2010*; *Wohlgemuth et al., 2016*) and PDB Ligand (*Shin, 2004*), and toxicological information, such as the EPA CompTox Dashboard (*Williams et al., 2017*). Additionally, these items contain links to compound classes, disease indications, pharmaceutical products, and protein targets.

**Pathways**: Wikidata has items for almost three thousand human biological pathways, primarily from two established public pathway repositories: Reactome (*Fabregat et al., 2018*) and WikiPathways (*Slenter et al., 2018*). The full details of the different pathways remain with the respective primary sources. Our bots enter data for Wikidata properties such as pathway name, identifier, organism, and the list of component genes, proteins, and chemical compounds. Properties for contributing authors (via ORCID properties; *Sprague, 2017*), descriptions and ontology annotations are also being added for Wikidata pathway entries.

**Diseases**: Wikidata has items for over 16 thousand diseases, the majority of which were created based on imports from the Human Disease Ontology (*Schriml et al., 2019*), with additional disease terms added from the Monarch Disease Ontology (*Mungall et al., 2017*). Disease attributes include medical classifications, symptoms, relevant drugs, as well as subclass relationships to higher-level disease categories. In instances where the Human Disease Ontology specifies a related anatomic region and/or a causative organism (for infectious diseases), corresponding statements are also added.

**References**: Whenever practical, the provenance of each statement added to Wikidata was also added in a structured format. References are part of the core data model for a Wikidata statement. References can either cite the primary resource from which the statement was retrieved (including details like version number of the resource), or they can link to a Wikidata item corresponding to a publication as provided by a primary resource (as an extension of the WikiCite project; *Ayers et al., 2019*), or both. Wikidata contains over 20 million items corresponding to publications across many domain areas, including a heavy emphasis on biomedical journal articles.

### Bot automation

To programmatically upload biomedical knowledge to Wikidata, we developed a series of computer programs, or bots. Bot development began by reaching a consensus on data modeling with the Wikidata community, particularly the Molecular Biology WikiProject. We then coded each bot to retrieve, transform, normalize and upload data from a primary resource to Wikidata via the Wikidata application programming interface (API).

We generalized the common code modules into a Python library, called Wikidata Integrator (WDI), to simplify the process of creating Wikidata bots (https://github.com/SuLab/WikidataIntegrator; archived at *Burgstaller-Muehlbacher et al., 2020*). Relative to accessing the API directly, WDI has convenient features that improve the bot development experience. These features include the creation of items for scientific articles as references, basic detection of data model conflicts, automated detection of items needing update, detailed logging and error handling, and detection and preservation of conflicting human edits.

Just as important as the initial data upload is the synchronization of updates between the primary sources and Wikidata. We utilized Jenkins, an open-source automation server, to automate all our Wikidata bots. This system allows for flexible scheduling, job tracking, dependency management, and automated logging and notification. Bots are either run on a predefined schedule (for continuously updated resources) or when new versions of original databases are released.

## Applications of Wikidata

Translating between identifiers from different databases is one of the most common operations in bioinformatics analyses. Unfortunately, these translations are most often done by bespoke scripts and based on entity-specific mapping tables. These translation scripts are repetitively and redundantly written across our community and are rarely kept up to date, nor integrated in a reusable fashion.

An identifier translation service is a simple and straightforward application of the biomedical content in Wikidata. Based on mapping tables that have been imported, Wikidata items can be mapped to databases that are both widely- and rarely-used in the life sciences community. Because all these mappings are stored in a centralized database and use a systematic data model, generic and reusable translation scripts can easily be written (*Figure 2*). These scripts can be used as a foundation for more complex Wikidata queries, or the results can be

```
SELECT * WHERE {
  values ?symbol {"CDK2" "AKT1" "RORA" "VEGFA" "COL2A1" "NGLY1"} .
   ?gene wdt:P353 ?symbol .
   ?gene wdt:P351 ?entrez .
}

SELECT * WHERE {
  values ?rxnorm {"327361" "301542" "10582" "284924"} .
   ?compound wdt:P3345 ?rxnorm .
   ?compound wdt:P2115 ?ndfrt .
}
```

Input ID type          Input IDs

Output ID type

**Figure 2.** Generalizable SPARQL template for identifier translation. SPARQL is the primary query language for accessing Wikidata content. These simple SPARQL examples show how identifiers of any biological type can easily be translated using SPARQL queries. The top query demonstrates the translation of a small list of gene symbols (wdt:P353) to Entrez Gene IDs (wdt:P351), while the bottom example shows conversion of RxNorm concept IDs (wdt:P3345) to NDF-RT IDs (wdt:P2115). These queries can be submitted to the Wikidata Query Service (WDQS; https://query.wikidata.org/) to get real-time results. Translation to and from a wide variety of identifier types can be performed using slight modifications on these templates, and relatively simple extensions of these queries can filter mappings based on the statement references and/or qualifiers. A full list of Wikidata properties can be found at https://www.wikidata.org/wiki/Special:ListProperties. Note that for translating a large number of identifiers, it is often more efficient to perform a SPARQL query to retrieve all mappings and then perform additional filtering locally.

downloaded and used as part of larger scripts or analyses.

There are a number of other tools that are also aimed at solving the identifier translation use case, including the BioThings APIs (*Xin et al., 2018*), BridgeDb (*van Iersel et al., 2010*), Bio-Mart (*Smedley et al., 2015*), UMLS (*Bodenreider, 2004*), and NCI Thesaurus (*de Coronado et al., 2009*). Relative to these tools, Wikidata distinguishes itself with a unique combination of the following: an almost limitless scope including all entities in biology, chemistry, and medicine; a data model that can represent exact, broader, and narrow matches between items in different identifier namespaces (beyond semantically imprecise 'cross-references'); programmatic access through web services with a track record of high performance and high availability.

Moreover, Wikidata is also unique as it is the only tool that allows real-time community editing. So while Wikidata is certainly not complete with respect to identifier mappings, it can be continually improved independent of any centralized effort or curation authority. As a database of assertions and not of absolute truth, Wikidata is able to represent conflicting information (with provenance) when, for example, different curation authorities produce different mappings between entities. (However, as with any bioinformatics integration exercise, harmonization of cross-references between resources

can include relationships other than 'exact match'. These instances can lead to Wikidata statements that are not explicitly declared, but rather the result of transitive inference.)

### Integrative Queries

Wikidata contains a much broader set of information than just identifier cross-references. Having biomedical data in one centralized data resource facilitates powerful integrative queries that span multiple domain areas and data sources. Performing these integrative queries through Wikidata obviates the need to perform many time-consuming and error-prone data integration steps.

As an example, consider a pulmonologist who is interested in identifying candidate chemical compounds for testing in disease models (schematically illustrated in *Figure 3*). They may start by identifying genes with a genetic association to any respiratory disease, with a particular interest in genes that encode membrane-bound proteins (for ease in cell sorting). They may then look for chemical compounds that either directly inhibit those proteins, or finding none, compounds that inhibit another protein in the same pathway. Because they have collaborators with relevant expertise, they may specifically filter for proteins containing a serine-threonine kinase domain.

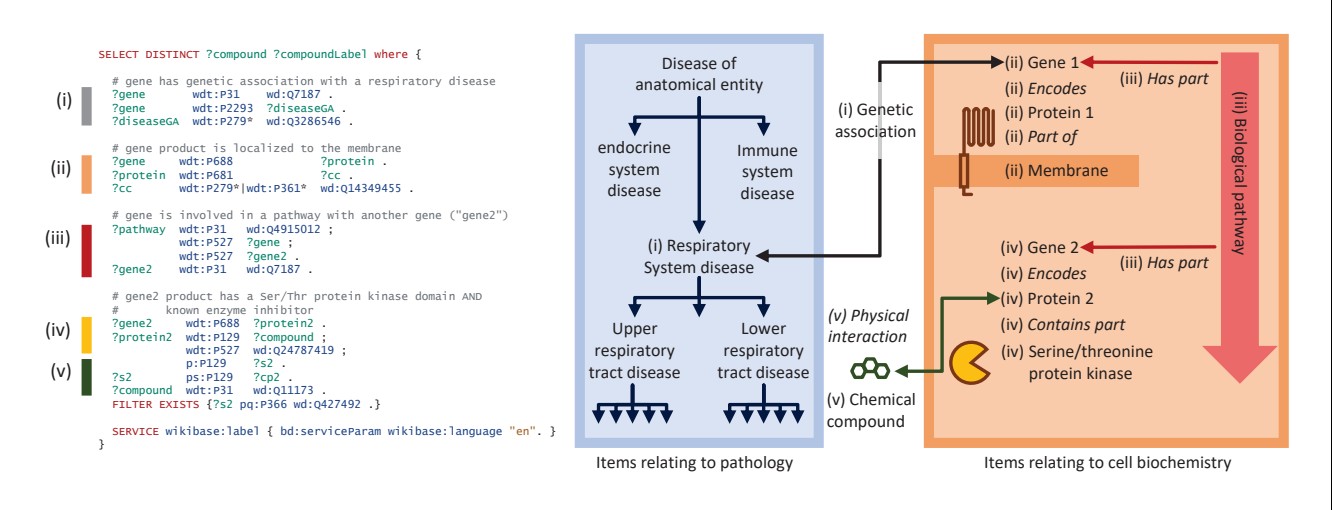

**Figure 3.** A representative SPARQL query that integrates data from multiple data resources and annotation types. This example integrative query incorporates data on genetic associations to disease, Gene Ontology annotations for cellular compartment, protein target information for compounds, pathway data, and protein domain information. Specifically, this query (depicted schematically at right) retrieves genes that are (i) associated with a respiratory system disease, (ii) that encode a membrane-bound protein, and (iii) that sit within the same biochemical pathway as (iv) a second gene encoding a protein with a serine-threonine kinase domain and (v) a known inhibitor, and reports a list of those inhibitors. Aspects related to Disease Ontology in blue; aspects related to biochemistry in red/orange; aspects related to chemistry in green. Properties are shown in italics. Real-time query results can be viewed at https://w.wiki/6pZ.

Almost any competent informatician can perform the query described above by integrating cell localization data from Gene Ontology annotations, genetic associations from GWAS Catalog, disease subclass relationships from the Human Disease Ontology, pathway data from WikiPathways and Reactome, compound targets from the IUPHAR Guide to Pharmacology, and protein domain information from InterPro. However, actually performing this data integration is a time-consuming and error-prone process. At the time of publication of this manuscript, this Wikidata query completed in less than 10 s and reported 31 unique compounds. Importantly, the results of that query will always be up-to-date with the latest information in Wikidata.

This query, and other example SPARQL queries that take advantage of the rich, heterogeneous knowledge network in Wikidata are available at https://www.wikidata.org/wiki/User:ProteinBoxBot/SPARQL_Examples. That page additionally demonstrates federated SPARQL queries that perform complex queries across other biomedical SPARQL endpoints. Federated queries are useful for accessing data that cannot be included in Wikidata directly due to limitations in size, scope, or licensing.

## Crowdsourced curation

Ontologies are essential resources for structuring biomedical knowledge. However, even after the initial effort in creating an ontology is finalized, significant resources must be devoted to maintenance and further development. These tasks include cataloging cross references to other ontologies and vocabularies, and modifying the ontology as current knowledge evolves. Community curation has been explored in a variety of tasks in ontology curation and annotation (see, for example, *Bunt et al., 2012*; *Gil et al., 2017*; *Putman et al., 2019*; *Putman et al., 2017*; *Wang et al., 2016*). While community curation offers the potential of distributing these responsibilities over a wider set of scientists, it also has the potential to introduce errors and inconsistencies.

Here, we examined how a crowd-based curation model through Wikidata works in practice. Specifically, we designed a hybrid system that combines the aggregated community effort of many individuals with the reliability of expert curation. First, we created a system to monitor, filter, and prioritize changes made by Wikidata contributors to items in the Human Disease Ontology. We initially seeded Wikidata with disease items from the Disease Ontology (DO) starting in late 2015. Beginning in 2018, we

compared the disease data in Wikidata to the most current DO release on a monthly basis.

In our first comparison between Wikidata and the official DO release, we found that Wikidata users added a total of 2030 new cross references to GARD (*Lewis et al., 2017*) and MeSH (https://www.nlm.nih.gov/mesh/meshhome.html). These cross references were primarily added by a small handful of users through a web interface focused on identifier mapping (*Manske, 2020*). Each cross reference was manually reviewed by DO expert curators, and 2007 of these mappings (98.9%) were deemed correct and therefore added to the ensuing DO release. 771 of the proposed mappings could not be easily validated using simple string matching, and 754 (97.8%) of these were ultimately accepted into DO. Each subsequent monthly report included a smaller number of added cross references to GARD and MeSH, as well as ORDO (*Maiella et al., 2018*), and OMIM (*Amberger and Hamosh, 2017*; *McKusick, 2007*), and these entries were incorporated after expert review at a high approval rate (>90%).

Addition of identifier mappings represents the most common community contribution, and likely the most accessible crowdsourcing task. However, Wikidata users also suggested numerous refinements to the ontology structure, including changes to the subclass relationships and the addition of new disease terms. These structural changes were more nuanced and therefore rarely incorporated into DO releases with no modifications. Nevertheless, they often prompted further review and refinement by DO curators in specific subsections of the ontology.

The Wikidata crowdsourcing curation model is generalizable to any other external resource that is automatically synced to Wikidata. The code to detect changes and assemble reports is tracked online at https://github.com/SuLab/scheduled-bots (archived at *Stupp et al., 2020*) and can easily be adapted to other domain areas. This approach offers a novel solution for integrating new knowledge into a biomedical ontology through distributed crowdsourcing while preserving control over the expert curation process. Incorporation into Wikidata also enhances exposure and visibility of the resource by engaging a broader community of users, curators, tools, and services.

### Interactive pathway pages
In addition to its use as a repository for data, we explored the use of Wikidata as a primary access and visualization endpoint for pathway data. We used Scholia, a web app for displaying scholarly profiles for a variety of Wikidata entries, including individual researchers, research topics, chemicals, and proteins (*Nielsen et al., 2017*). Scholia provides a more user-friendly view of Wikidata content with context and interactivity that is tailored to the entity type.

We contributed a Scholia profile template specifically for biological pathways (*Scholia, 2019*). In addition to essential items such as title and description, these pathway pages include an interactive view of the pathway diagram collectively drawn by contributing authors. The WikiPathways identifier property in Wikidata informs the Scholia template to source a *pathway-viewer* widget from Toolforge (https://tools.wmflabs.org/admin/tool/pathway-viewer) that in turn retrieves the corresponding interactive pathway image. Embedded into the Scholia pathway page, the widget provides pan and zoom, plus links to gene, protein and chemical Scholia pages for every clickable molecule on the pathway diagram see, for example, *Scholia (2019)*. Each pathway page also includes information about the pathway authors. The Scholia template also generates a participants table that shows the genes, proteins, metabolites, and chemical compounds that play a role in the pathway, as well as citation information in both tabular and chart formats.

With Scholia template views of Wikidata, we were able to generate interactive pathway pages with comparable content and functionality to that of dedicated pathway databases. Wikidata provides a powerful interface to access these biological pathway data in the context of other biomedical knowledge, and Scholia templates provide rich, dynamic views of Wikidata that are relatively simple to develop and maintain.

### Phenotype based disease diagnosis
Phenomizer is a web application that suggests clinical diagnoses based on an array of patient phenotypes (*Köhler et al., 2009*). On the back end, the latest version of Phenomizer uses BOQA, an algorithm that uses ontological structure in a Bayesian network (*Bauer et al., 2012*). For phenotype-based disease diagnosis, BOQA takes as input a list of phenotypes (using the Human Phenotype Ontology [HPO; *Köhler et al., 2017*]) and an association file between phenotypes and diseases. BOQA then suggests disease diagnoses based on semantic similarity (*Köhler et al., 2009*). Here, we studied whether phenotype-disease associations from

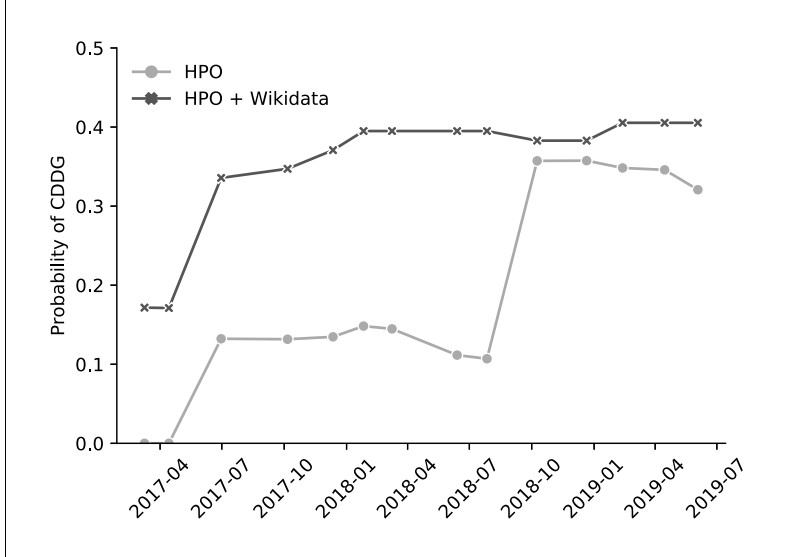

**Figure 4.** BOQA analysis of suspected cases of the disease Congenital Disorder of Deglycosylation (CDDG). We used an algorithm called BOQA to rank potential diagnoses based on clinical phenotypes. Here, clinical phenotypes from two cases of suspected CDDG patients were extracted from a published case report (*Caglayan et al., 2015*). These phenotypes were run through BOQA using phenotype-disease annotations from the Human Phenotype Ontology (HPO) alone, or from a combination of HPO and Wikidata. This analysis was tested using several versions of disease-phenotype annotations (shown along the x-axis). The probability score for CDDG is reported on the y-axis. These results demonstrate that the inclusion of Wikidata-based disease-phenotype annotations would have significantly improved the diagnosis predictions from BOQA at earlier time points prior to their official inclusion in the HPO annotation file. Details of this analysis can be found at https://github.com/SuLab/Wikidata-phenomizer (archived at *Tu et al., 2020*).

Wikidata could improve BOQA's ability to make differential diagnoses for certain sets of phenotypes. We modified the BOQA codebase to accept arbitrary inputs and to be able to run from the command line (code available at https://github.com/SuLab/boqa; archived at *Köhler and Stupp, 2020*) and also wrote a script to extract and incorporate the phenotype-disease annotations in Wikidata (code available at https://github.com/SuLab/Wikidata-phenomizer; archived at *Tu et al., 2020*).

As of September 2019, there were 273 phenotype-disease associations in Wikidata that were not in the HPO's annotation file (which contained a total of 172,760 associations). Based on parallel biocuration work by our team, many of these new associations were related to the disease Congenital Disorder of Deglycosylation (CDDG; also known as NGLY-1 deficiency) based on two papers describing patient phenotypes (*Enns et al., 2014*; *Lam et al., 2017*). To see if the Wikidata-sourced annotations improved the ability of BOQA to diagnose CDDG, we ran our modified version using the phenotypes taken

from a third publication describing two siblings with suspected cases of CDDG (*Caglayan et al., 2015*). Using these phenotypes and the annotation file supplemented with Wikidata-derived associations, BOQA returned a much stronger semantic similarity to CDDG relative to the HPO annotation file alone (*Figure 4*). Analyses with the combined annotation file reported CDDG as the top result for each of the past 14 releases of the HPO annotation file, whereas CDDG was never the top result when run without the Wikidata-derived annotations.

This result demonstrated an example scenario in which Wikidata-derived annotations could be a useful complement to expert curation. This example was specifically chosen to illustrate a favorable case, and the benefit of Wikidata would likely not currently generalize to a random sampling of other diseases. Nevertheless, we believe that this proof-of-concept demonstrates the value of the crowd-based Wikidata model and may motivate further community contributions.

### Drug repurposing

The mining of graphs for latent edges has been an area of interest in a variety of contexts from predicting friend relationships in social media platforms to suggesting movies based on past viewing history. A number of groups have explored the mining of knowledge graphs to reveal biomedical insights, with the open source Rephetio effort for drug repurposing as one example (*Himmelstein et al., 2017*). Rephetio uses logistic regression, with features based on graph metapaths, to predict drug repurposing candidates.

The knowledge graph that served as the foundation for Rephetio was manually assembled from many different resources into a heterogeneous knowledge network. Here, we explored whether the Rephetio algorithm could successfully predict drug indications on the Wikidata knowledge graph. Based on the class diagram in *Figure 1*, we extracted a biomedically-focused subgraph of Wikidata with 19 node types and 41 edge types. We performed five-fold cross validation on drug indications within Wikidata and found that Rephetio substantially enriched the true indications in the hold-out set. We then downloaded historical Wikidata versions from 2017 and 2018 and observed marked improvements in performance over time (*Figure 5*). We also performed this analysis using an external test set based on Drug Central, which showed a similar improvement in

Rephetio results over time (*Figure 5—figure supplement 1*).

This analysis demonstrates the value of a community-maintained, centralized knowledge base to which many researchers are contributing. It suggests that scientific analyses based on Wikidata may continually improve irrespective of any changes to the underlying algorithms, but simply based on progress in curating knowledge through the distributed, and largely uncoordinated efforts of the Wikidata community.

## Outlook

We believe that the design of Wikidata is well-aligned with the FAIR data principles.

**Findable**: Wikidata items are assigned globally unique identifiers with direct cross-links into the massive online ecosystem of Wikipedias. Wikidata also has broad visibility within the Linked Data community and is listed in the life science registries FAIRsharing (https://fairsharing.org/; *Sansone et al., 2019*) and Identifiers.org (*Wimalaratne et al., 2018*). Wikidata has already attracted a robust, global community of contributors and consumers.

**Accessible**: Wikidata provides access to its underlying knowledge graph via both an online graphical user interface and an API, and access includes both read- and write-privileges. Wikidata provides database dumps at least weekly (https://www.wikidata.org/wiki/Wikidata:Database_download), ensuring the long-term accessibility of the Wikidata knowledge graph independent of the organization and web application. Finally, Wikidata is also natively multilingual.

**Interoperable**: Wikidata items are extensively cross-linked to other biomedical resources using Universal Resource Identifiers (URIs), which unambiguously anchor these concepts in the Linked Open Data cloud (*Jacobsen et al., 2018*). Wikidata is also available in many standard formats in computer programming and knowledge management, including JSON, XML, and RDF.

**Reusable**: Data provenance is directly tracked in the reference section of the Wikidata statement model. The Wikidata knowledge graph is released under the Creative Commons Zero (CC0) Public Domain Declaration, which explicitly declares that there are no restrictions on downstream reuse and redistribution.

The open data licensing of Wikidata is particularly notable. The use of data licenses in biomedical research has rapidly proliferated, presumably in an effort to protect intellectual property and/or justify long-term grant funding (see, for example, *Reiser et al., 2016*). However, even seemingly innocuous license terms (like requirements for attribution) still impose legal requirements and therefore expose consumers to legal liability. This liability is especially problematic for data integration efforts, in which the license terms of all resources (dozens or hundreds or more) must be independently tracked and satisfied (a phenomenon referred to as 'license stacking'). Because it is released under CC0, Wikidata can be freely and openly used in any other resource without any restriction. This freedom greatly simplifies and encourages downstream use, albeit at the cost of not being able to incorporate ontologies or datasets with more restrictive licensing.

In addition to simplifying data licensing, Wikidata offers significant advantages in centralizing the data harmonization process. Consider the use case of trying to get a comprehensive list of disease indications for the drug bupropion. The National Drug File – Reference Terminology (NDF-RT) reported that bupropion may treat nicotine dependence and attention deficit hyperactivity disorder, the Inxight database listed major depressive disorder, and the FDA Adverse Event Reporting System (FAERS) listed anxiety and bipolar disorder. While no single database listed all these indications, Wikidata provided an integrated view that enabled seamless query and access across resources. Integrating drug indication data from these individual data resources was not a trivial process. Both Inxight and NDF-RT mint their own identifiers for both drugs and diseases. FAERS uses Medical Dictionary for Regulatory Activities (MedDRA) names for diseases and free-text names for drugs (*Stupp and Su, 2018*). By harmonizing and integrating all resources in the context of Wikidata, we ensure that those data are immediately usable by others without having to repeat the normalization process. Moreover, by harmonizing data at the time of data loading, consumers of that data do not need to perform the repetitive and redundant work at the point of querying and analysis.

As the biomedical data within Wikidata continues to grow, we believe that its unencumbered use will spur the development of many new innovative tools and analyses. These innovations will undoubtedly include the machine learning-based mining of the knowledge graph to predict new relationships (also referred to as knowledge graph reasoning; *Das et al., 2017*; *Lin et al., 2018*; *Xiong et al., 2017*).

For those who subscribe to this vision for cultivating a FAIR and open graph of biomedical

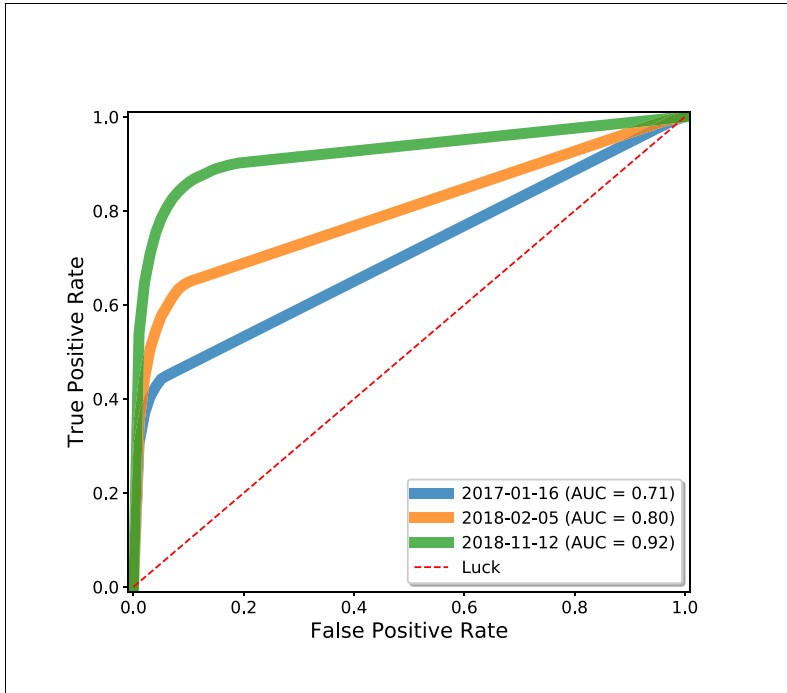

**Figure 5.** Drug repurposing using the Wikidata knowledge graph. We analyzed three snapshots of Wikidata using Rephetio, a graph-based algorithm for predicting drug repurposing candidates (*Himmelstein et al., 2017*). We evaluated the performance of the Rephetio algorithm on three historical versions of the Wikidata knowledge graph, quantified based on the area under the receiver operator characteristic curve (AUC). This analysis demonstrated that the performance of Rephetio in drug repurposing improved over time based only on improvements to the underlying knowledge graph. Details of this analysis can be found at https://github.com/SuLab/WD-rephetio-analysis (archived at *Mayers and Su, 2020*).

The online version of this article includes the following figure supplement(s) for figure 5:

**Figure supplement 1.** Drug repurposing using the Wikidata knowledge graph, evaluated using an external test set.

knowledge, there are two simple ways to contribute to Wikidata. First, owners of data resources can release their data using the CC0 declaration. Because Wikidata is released under CC0, it also means that all data imported in Wikidata must also use CC0-compatible terms (e.g., be in the public domain). For resources that currently use a restrictive data license primarily for the purposes of enforcing attribution or citation, we encourage the transition to CC0 (+BY), a model that "move[s] the attribution from the legal realm into the social or ethical realm by pairing a permissive license with a strong moral entreaty' (*Cohen, 2013*). For resources that must retain data license restrictions, consider releasing a subset of data or older versions of data using CC0. Many biomedical resources were created under or transitioned to CC0 (in part or in full) in recent years ,

including the Disease Ontology (*Schriml et al., 2019*), Pfam (*El-Gebali et al., 2019*), Bgee (*Bastian et al., 2008*), WikiPathways (*Slenter et al., 2018*), Reactome (*Fabregat et al., 2018*), ECO (*Chibucos et al., 2014*), and CIViC (*Griffith et al., 2017*).

Second, informaticians can contribute to Wikidata by adding the results of data parsing and integration efforts to Wikidata as, for example, new Wikidata items, statements, or references. Currently, the useful lifespan of data integration code typically does not extend beyond the immediate project-specific use. As a result, that same data integration process is likely performed repetitively and redundantly by other informaticians elsewhere. If every informatician contributed the output of their effort to Wikidata, the resulting knowledge graph would be far more useful than the stand-alone contribution of any single individual, and it would continually improve in both breadth and depth over time. Indeed, the growth of biomedical data in Wikidata is driven not by any centralized or coordinated process, but rather the aggregated effort and priorities of Wikidata contributors themselves.

FAIR and open access to the sum total of biomedical knowledge will improve the efficiency of biomedical research. Capturing that information in a centralized knowledge graph is useful for experimental researchers, informatics tool developers and biomedical data scientists. As a continuously-updated and collaboratively-maintained community resource, we believe that Wikidata has made significant strides toward achieving this ambitious goal.

## Acknowledgements
The authors thank the thousands of Wikidata contributors for curating knowledge, both directly related and unrelated to this work, much of which has been organized under the WikiProjects for Molecular Biology, Chemistry and Medicine. The authors also thank the Wikimedia Foundation for financially supporting Wikidata, and many developers and administrators for maintaining Wikidata as a community resource.

**Andra Waagmeester** is at Micelio, Antwerp, Belgium
https://orcid.org/0000-0001-9773-4008

**Gregory Stupp** is in the Department of Integrative Structural and Computational Biology, The Scripps Research Institute, La Jolla, CA, United States
https://orcid.org/0000-0002-0644-7212

**Sebastian Burgstaller-Muehlbacher** is in the Center for Integrative Bioinformatics Vienna, Max Perutz

Laboratories, University of Vienna and Medical University of Vienna, Vienna, Austria
https://orcid.org/0000-0003-4640-3510

**Benjamin M Good** is in the Department of Integrative Structural and Computational Biology, The Scripps Research Institute, La Jolla, CA, United States
https://orcid.org/0000-0002-7334-7852

**Malachi Griffith** is in the McDonnell Genome Institute, Washington University School of Medicine, St. Louis, MO, United States
https://orcid.org/0000-0002-6388-446X

**Obi L Griffith** is in the McDonnell Genome Institute, Washington University School of Medicine, St. Louis, MO, United States
https://orcid.org/0000-0002-0843-4271

**Kristina Hanspers** is in the Institute of Data Science and Biotechnology, Gladstone Institutes, San Francisco, CA, United States
https://orcid.org/0000-0001-5410-599X

**Henning Hermjakob** is at the European Bioinformatics Institute (EMBL-EBI), Hinxton, United Kingdom
https://orcid.org/0000-0001-8479-0262

**Toby S Hudson** is in the School of Chemistry, University of Sydney, Sydney, Australia
https://orcid.org/0000-0002-3348-3622

**Kevin Hybiske** is in the Division of Allergy and Infectious Diseases, Department of Medicine, University of Washington, Seattle, WA, United States
https://orcid.org/0000-0002-2967-3079

**Sarah M Keating** is at the European Bioinformatics Institute (EMBL-EBI), Hinxton, United Kingdom
https://orcid.org/0000-0002-3356-3542

**Magnus Manske** is at the Wellcome Trust Sanger Institute, Hinxton, United Kingdom
https://orcid.org/0000-0001-5916-0947

**Michael Mayers** is in the Department of Integrative Structural and Computational Biology, The Scripps Research Institute, La Jolla, CA, United States
https://orcid.org/0000-0002-7792-0150

**Daniel Mietchen** is in the School of Data Science, University of Virginia, Charlottesville, VA, United States
https://orcid.org/0000-0001-9488-1870

**Elvira Mitraka** is in the University of Maryland School of Medicine, Baltimore, MD, United States
https://orcid.org/0000-0003-0719-3485

**Alexander R Pico** is in the Institute of Data Science and Biotechnology, Gladstone Institutes, San Francisco, CA, United States
https://orcid.org/0000-0001-5706-2163

**Timothy Putman** is in the Department of Integrative Structural and Computational Biology, The Scripps Research Institute, La Jolla, CA, United States
https://orcid.org/0000-0002-4291-0737

**Anders Riutta** is in the Institute of Data Science and Biotechnology, Gladstone Institutes, San Francisco, CA, United States
https://orcid.org/0000-0002-4693-0591

**Núria Queralt-Rosinach** is in the Department of Integrative Structural and Computational Biology, The Scripps Research Institute, La Jolla, CA, United States
https://orcid.org/0000-0003-0169-8159

**Lynn M Schriml** is in the University of Maryland School of Medicine, Baltimore, MD, United States
https://orcid.org/0000-0001-8910-9851

**Thomas Shafee** is in the Department of Animal Plant and Soil Sciences, La Trobe University, Melbourne, Australia
https://orcid.org/0000-0002-2298-7593

**Denise Slenter** is in the Department of Bioinformatics-BiGCaT, NUTRIM, Maastricht University, Maastricht, Netherlands
https://orcid.org/0000-0001-8449-1318

**Ralf Stephan** is a retired researcher based in Berlin, Germany
https://orcid.org/0000-0002-4650-631X

**Katherine Thornton** is at Yale University Library, Yale University, New Haven, CT, United States
https://orcid.org/0000-0002-4499-0451

**Ginger Tsueng** is in the Department of Integrative Structural and Computational Biology, The Scripps Research Institute, La Jolla, CA, United States
https://orcid.org/0000-0001-9536-9115

**Roger Tu** is in the Department of Integrative Structural and Computational Biology, The Scripps Research Institute, La Jolla, CA, United States
https://orcid.org/0000-0002-7899-1604

**Sabah Ul-Hasan** is in the Department of Integrative Structural and Computational Biology, The Scripps Research Institute, La Jolla, CA, United States
https://orcid.org/0000-0001-6334-452X

**Egon Willighagen** is in the Department of Bioinformatics-BiGCaT, NUTRIM, Maastricht University, Maastricht, Netherlands
https://orcid.org/0000-0001-7542-0286

**Chunlei Wu** is in the Department of Integrative Structural and Computational Biology, The Scripps Research Institute, La Jolla, CA, United States
https://orcid.org/0000-0002-2629-6124

**Andrew I Su** is in the Department of Integrative Structural and Computational Biology, The Scripps Research Institute, La Jolla, CA, United States
asu@scripps.edu
https://orcid.org/0000-0002-9859-4104

*Author contributions:* Andra Waagmeester, Conceptualization, Data curation, Software, Formal analysis, Validation, Visualization, Writing - review and editing; Gregory Stupp, Conceptualization, Data curation, Software, Formal analysis, Validation, Visualization, Writing - original draft; Sebastian Burgstaller-Muehlbacher, Conceptualization, Data curation, Software; Benjamin M Good, Conceptualization, Data curation, Software, Supervision, Project administration; Malachi Griffith, Kevin Hybiske, Data curation, Funding acquisition; Obi L Griffith, Data curation, Funding acquisition, Writing - review and editing; Kristina Hanspers, Sarah M

Keating, Magnus Manske, Timothy Putman, Anders Riutta, Nuria Queralt-Rosinach, Denise Slenter, Ginger Tsueng, Sabah Ul-Hasan, Egon Willighagen, Data curation, Software; Henning Hermjakob, Lynn M Schriml, Data curation, Supervision, Funding acquisition; Toby S Hudson, Elvira Mitraka, Ralf Stephan, Data curation, Validation; Michael Mayers, Data curation, Software, Formal analysis, Visualization; Daniel Mietchen, Data curation, Validation, Writing - review and editing; Alexander R Pico, Data curation, Supervision, Writing - original draft, Writing - review and editing; Thomas Shafee, Data curation, Visualization, Writing - review and editing; Katherine Thornton, Software, Validation; Roger Tu, Software, Formal analysis, Visualization; Chunlei Wu, Data curation, Software, Supervision, Funding acquisition; Andrew I Su, Conceptualization, Formal analysis, Supervision, Funding acquisition, Validation, Writing - original draft, Project administration, Writing - review and editing

*Competing interests:* The authors declare that no competing interests exist.

### Funding

| Funder | Grant reference number | Author |
|---|---|---|
| National Institute of General Medical Sciences | R01 GM089820 | Andrew I Su |
| National Institute of General Medical Sciences | U54 GM114833 | Henning Hermjakob Andrew I Su |
| National Institute of General Medical Sciences | R01 GM100039 | Alexander R Pico |
| National Human Genome Research Institute | R00HG007940 | Malachi Griffith |
| National Cancer Institute | U24CA237719 | Malachi Griffith |
| V Foundation for Cancer Research | V2018-007 | Malachi Griffith |
| National Institute of Allergy and Infectious Diseases | R01 AI126785 | Kevin Hybiske |
| National Center for Advancing Translational Sciences | UL1 TR002550 | Andrew I Su |

The funders had no role in study design, data collection and interpretation, or the decision to submit the work for publication.

### Decision letter and Author response

Decision letter https://doi.org/10.7554/eLife.52614.sa1
Author response https://doi.org/10.7554/eLife.52614.sa2

## Additional files

### Data availability

Links to all data and code used in this manuscript have been provided.

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
