## [Decision Letter]

Thank you for submitting your article "Wikidata as a FAIR knowledge graph for the life sciences" for consideration by *eLife*. Your article has been reviewed by two peer reviewers, and the evaluation has been overseen by the *eLife* Features Editor (Peter Rodgers). The following individuals involved in review of your submission have agreed to reveal their identity: Chris Mungall (Reviewer #3).

The reviewers have discussed the reviews with one another and the Features Editor has drafted this decision to help you prepare a revised submission. We hope you will be able to submit the revised version within two months.

SUMMARY

The manuscript describes the life sciences component of the Wikidata knowledge base, which combines multiple disaggregated knowledge/databases into a single source enabling integrated querying. The authors describe the process for keeping Wikidata up to date, and they describe compelling use cases showing the power. Although there are many efforts in the life sciences to create integrated knowledge bases / knowledge graphs, Wikidata is unique in the breadth, scope, and in the community/crowdsourcing aspects. It is an increasingly important resource in the biomedical landscape, and the manuscript provides a clear description of the authors' significant efforts in building this component of the resource. However, the manuscript would benefit from addressing a number of issues in greater detail - see below.

ESSENTIAL REVISIONS:

1. An argument is made that Wikidata combines centralized and distributed approaches. This is an interesting concept. It would be important to know the relative importance of those two approaches. In other words, what proportion of edits is centralized (bots) and what proportion is distributed (contributors). Authors repeatedly refer to the fact that anyone is empowered to add new content, but how much of this potential is realized, and how this compares to other Wiki projects, e.g., Wikipedia?

2. Performance metrics and measurable assessment are often missing from the description of the applications. For example, when the process of translation of identifiers from various databases is described, no information is provided if the translation is done deterministically or probabilistically, how it is done algorithmically, how the performance can be evaluated (AUC ROC?), how Wikidata compares to other systems (e.g., UMLS). It is admirable that the authors admit that "Wikidata is certainly not complete with respect to identifier mappings...", but readers would benefit from knowing more about the proportion of mappings available and those not, and how this differs across databases integrated within Wikidata.

3. In such a broad overview, including multiple use cases, some selectivity in the presentation of examples is probably inevitable. This is understandable. Yet one can make those choices more informed, and offer, in addition, examples less favorable to the system (Wikidata), and in this way provide a balanced assessment. For example, on pages 10-11, for the purpose of disease diagnosis, annotations from the Human Phenotype Ontology (HPO) were compared to combined HPO + Wikidata annotations. However, comparison to only one set of annotations with respect to only one disease could be uninformative. At least, performance should be evaluated against a set of different randomly selected diseases.

4. An argument is made that the "Wikidata is among the most FAIR biomedical resources available". What are the other resources, and how Wikidata compares to those resources with respect to the four FAIR criteria?

5. The FAIR principles do not address the issue of completeness, as they were designed for datasets which are inherently self-contained and presumably complete with respect to the experiment/study. For biomedical knowledge bases (KBs), especially warehouse-style, a key question is: how complete is the KB? How often will my queries have false negatives? (Of course, KBs are always incomplete due to both knowledge and curation gaps. However, it is still important to provide users a sense of the areas in which there are major gaps, for example due to inability to integrate a key resource due to licensing).

An example is on figure 1: there are only 636 disease to symptom/phenotype associations. This is a very small amount, when compared to reference resources like the Human Phenotype Ontology. This may have negative consequences for applications built around Wikidata, causing people to lose trust. Another example is in the subset of genes imported - from 200 species. What determines the set of species included? Are there performance implications in including all genes in all species?

Of course, this is a difficult challenge for any KB, and completeness is not expected. I would still recommend a section that acknowledges and addresses this, addressing questions such as:

- how is a user best able to determine whether a portion of Wikidata is complete enough for their use case

- what is the process for deciding what is included vs excluded?

6. Identifier merging details: The identifier mapping use case is well articulated, but some details are omitted. Some identifier mapping resources store pairwise mappings without privileging any one resource. Wikidata is more like UMLS, in which they mint a new concept ID for the unified concept, and map everything to this. The challenge with this scheme is deciding on the criteria for lumping and splitting. Many source mappings are not 1:1, which can lead to excessive merging when these are traversed transitively

- For bot-derived entities, what is the algorithm for doing this?

- Is one source (e.g. NCBI gene) taken as canonical?

Additionally, it would be useful to see a stricter comparison with sources like bridgedb, in terms of completeness and consistency.

7. Trustworthiness of curation: The strength of Wikidata is in the crowdsourcing of knowledge. While this can be scaled up more easily due less of a need for funded curators, the downside is that the information may be less accurate and reliable. The paper provides good evidence of reliability via the disease cross-referencing experiment, in which 99% of 2030 crowd-sourced mappings were reviewed and accepted.

I am a little skeptical of the generalizability of these results. Mappings are generally quite easy and can be done with reasonable reliability by an automated string-matching process (with some caveats). It would be useful to know more about how many different people contributed to the 2030 (was it one person running a script)?

Refinements to the ontology structure are harder, and it's not clear how often these were incorporated. The mapping results are still a nice example, but they need to be qualified more, and there should be more discussion on reliability.

8. Phenomizer analysis: It's not clear if there is circularity in the Phenomizer analysis:

- were there publications that incorporated the case reports that were then annotated? Was this controlled for?

- What happened on 2018-09 when the HPO-only semsim score jumps? Were the WD annotations incorporated into HPO? Or was this independent annotation? The subsequent drop is curious.

9. Some lines read as if written for a grant proposal, e.g., "Wikidata has a proven track record for leveraging...", "Wikimedia Foundation... has a long track record of developing and maintaining..." see Introduction. In many instances, Wikidata is presented as "unique". Please reword such sentences/passages.

CODE/DATA

The Phenomizer analysis can't be reproduced at the moment, as the curated case reports with HPO IDs are not made available. Additionally, the settings for Phenomizer should be made available, also Phenomizer provides p-values, it would be useful to see this in the analysis results.

---

## [Author Response]

We thank you and the reviewers for the helpful and detailed evaluation of our manuscript. We have made many changes in response to these comments, and believe the manuscript is substantially strengthened as a result. A point-by-point description of these changes is included below. We repeat the reviewers’ points here in italic, followed by our response in plain text.

SUMMARYThe manuscript describes the biomedical life sciences component of the Wikidata knowledge base, which combines multiple disaggregated knowledge/databases into a single source enabling integrated querying. The authors describe the process for keeping Wikidata up to date, and they describe compelling use cases showing the power. Although there are many efforts in the life sciences to create integrated knowledge bases / knowledge graphs, Wikidata is unique in the breadth, scope, and in the community/crowdsourcing aspects. It is an increasingly important resource in the biomedical landscape, and the manuscript provides a clear description of the authors' significant efforts in building this component of the resource. However, the manuscript would benefit from addressing a number of issues in greater detail - see below.ESSENTIAL REVISIONS:1. An argument is made that Wikidata combines centralized and distributed approaches. This is an interesting concept. It would be important to know the relative importance of those two approaches. In other words, what proportion of edits is centralized (bots) and what proportion is distributed (contributors). Authors repeatedly refer to the fact that anyone is empowered to add new content, but how much of this potential is realized, and how this compares to other Wiki projects, e.g., Wikipedia?

The Wikimedia Statistics tracker shows that over the last two years, 46% of all edits on Wikidata were attributed to normal "user" accounts while ~54% are attributed to accounts that are registered as bots, with the trend over the last year showing an increasing trend toward non-bot user edits ((Wikimedia Foundation, 2020); Figure 1-figure supplement 1 in revised manuscript). For comparison, that same site reports that English Wikipedia had 69% user edits and 14% bot edits (and an additional 17% anonymous edits).

In addition to the statistics above that demonstrate a balance between human and bot edits, the absolute number of editors is noteworthy. The number of "active editors" (five or more edits per month) has steadily grown from 10k to 12k over the last two years, with the number of bot accounts staying relatively stable over that time.

Finally, we also note that while Wikidata bots tend to focus on importing large-scale 'centralized' resources, the bot community itself is made up of the decentralized and distributed efforts of many bot developers.

These points have been clarified in the manuscript (including the addition of a new Figure 1-figure supplement 1).

2. Performance metrics and measurable assessment are often missing from the description of the applications. For example, when the process of translation of identifiers from various databases is described, no information is provided if the translation is done deterministically or probabilistically, how it is done algorithmically, how the performance can be evaluated (AUC ROC?), how Wikidata compares to other systems (e.g., UMLS). It is admirable that the authors admit that "Wikidata is certainly not complete with respect to identifier mappings...", but readers would benefit from knowing more about the proportion of mappings available and those not, and how this differs across databases integrated within Wikidata.

It is important to note that Wikidata is designed to be a database of assertions (with provenance) and not a database that attempts to resolve disagreements in search of absolute truth. Therefore, all identifier mappings in Wikidata are added deterministically – each Wikidata statement is an assertion from a database or curation authority. If different mapping resources disagree, then Wikidata can and should reflect that disagreement. Each consumer of Wikidata content is then free to add filters to prioritize or ignore specific mappings according to their own rules and biases. However, we also recognize that identifiers are a special class of statements because they establish the entities that are the subjects/objects of other statements. We provide more discussion of these topics below in response to point #6.

3. In such a broad overview, including multiple use cases, some selectivity in the presentation of examples is probably inevitable. This is understandable. Yet one can make those choices more informed, and offer, in addition, examples less favorable to the system (Wikidata), and in this way provide a balanced assessment. For example, on pages 10-11, for the purpose of disease diagnosis, annotations from the Human Phenotype Ontology (HPO) were compared to combined HPO + Wikidata annotations. However, comparison to only one set of annotations with respect to only one disease could be uninformative. At least, performance should be evaluated against a set of different randomly selected diseases.

Our goal was not to conclusively state that HPO + Wikidata is uniformly better for Phenomizer analyses. We acknowledge that the NGLY1 deficiency example was specifically selected to demonstrate a favorable case, and that the benefit of Wikidata would likely not currently generalize to a random sampling of other diseases. In this manuscript, we have attempted to balance examples in which Wikidata already has a rigorously-proven benefit to researchers, and proof-of-concept examples that demonstrate the value of the crowd-based Wikidata model and that motivate community contributions. Phenotype-based disease classification falls into the latter category. We have clarified this point in the manuscript.

4. An argument is made that the "Wikidata is among the most FAIR biomedical resources available". What are the other resources, and how Wikidata compares to those resources with respect to the four FAIR criteria?

Our intent is primarily to describe how Wikidata aligns with the FAIR data principles, and not to perform a comparative analysis of FAIRness relative to other biomedical resources. Therefore, we have reworded the text accordingly.

5. The FAIR principles do not address the issue of completeness, as they were designed for datasets which are inherently self-contained and presumably complete with respect to the experiment/study. For biomedical knowledge bases (KBs), especially warehouse-style, a key question is: how complete is the KB? How often will my queries have false negatives? (Of course, KBs are always incomplete due to both knowledge and curation gaps. However, it is still important to provide users a sense of the areas in which there are major gaps, for example due to inability to integrate a key resource due to licensing).An example is on figure 1: there are only 636 disease to symptom/phenotype associations. This is a very small amount, when compared to reference resources like the Human Phenotype Ontology. This may have negative consequences for applications built around Wikidata, causing people to lose trust. Another example is in the subset of genes imported - from 200 species. What determines the set of species included? Are there performance implications in including all genes in all species?Of course, this is a difficult challenge for any KB, and completeness is not expected. I would still recommend a section that acknowledges and addresses this, addressing questions such as:- how is a user best able to determine whether a portion of Wikidata is complete enough for their use case- what is the process for deciding what is included vs excluded?

There are many important issues raised in this point. Our overarching strategy for conveying the completeness is the class diagram in Figure 1. We believe that that Figure contains key information on the number of each entity type and the numbers of relationships between entity types. This information is relevant for each user to assess whether the appropriate data exist within Wikidata for their particular use case.

What is not included in Figure 1 are denominators – estimates of the total number of nodes/edges of a given type. Those denominators are extremely difficult to provide, since there are many reasonable ways to define them. For example, they could be based on resources that have been identified and screened but not yet loaded, based on resources that have been identified but are not suitable for Wikidata (e.g., for licensing reasons), based on an estimate of all structured knowledge (and/or unstructured knowledge) that is currently known, or based on an estimate of all knowledge that will eventually be known and discovered. We believe that the wide differences in how those denominators would be interpreted (and significant challenges in computing them) would counter any benefit to potential users. Instead, we have added a new supplemental table that shows the data sources that are cited as references for the most common properties. While this is not exactly the information that was requested by the reviewers, we believe this information will be more clearly interpretable.

A related question is raised here about how we prioritize resources for inclusion in Wikidata. In principle, we could write bots to load nearly any suitable data resources we come across. However, the bot writing process is actually a small proportion of the overall effort necessary to load a new resource. Each additional data resource requires a time-consuming data modeling process (in coordination with the broader Wikidata community and often the maintainers of the resource to be imported), adds overhead to our bot maintenance requirements, and introduces complexity in our data synchronization routines. Therefore, we avoid indiscriminate "stamp collecting" efforts in favor of targeted data loading driven by our own data-mining priorities. Accordingly, readers should feel empowered to import new data of interest to them, as that has been the primary mechanism for Wikidata's growth.

These points have been clarified in the manuscript.

6. Identifier merging details: The identifier mapping use case is well articulated, but some details are omitted. Some identifier mapping resources store pairwise mappings without privileging any one resource. Wikidata is more like UMLS, in which they mint a new concept ID for the unified concept, and map everything to this. The challenge with this scheme is deciding on the criteria for lumping and splitting. Many source mappings are not 1:1, which can lead to excessive merging when these are traversed transitively.- For bot-derived entities, what is the algorithm for doing this?- Is one source (e.g. NCBI gene) taken as canonical?

Wikidata bots generally do take one identifier as canonical (e.g., NCBI gene for genes, UniProt for proteins), but there is no requirement that other bots nor human editors use that same canonical identifier. Wikidata is based on a community process for harmonizing disconnected data resources. That means that any differing views on identifier mappings, like all statements on Wikidata items, are resolved via community discussion and consensus. Community input is solicited at many levels, starting at the approval process for bots to import large resources, all the way to discussions over individual statements on individual Wikidata items.

So while the process for resolving identifier mappings is the same as for any other Wikidata statements, we recognize that choices in identifier mappings have greater downstream implications (related to, for example, transitivity). However, the community-focused design of Wikidata means that there is no systematic scheme for making consistent lumping and splitting decisions. We recognize (and the original Wikidata creators recognized) that this design choice is not without its flaws, but also that it does have significant advantages with respect to usability at query time.

Finally, it is important to recognize that the issue of lumping and splitting is not specific to Wikidata, but one that exists across the biomedical knowledge management community. This issue has been exacerbated by the widespread use of semantically-imprecise cross references ('hasDbXref') in biomedical ontologies (as explained in this detailed blog post from Chris Mungall (Mungall, 2019)). So while we agree that this issue is not completely solved within Wikidata, we believe that the core issue lies further upstream in the knowledge management ecosystem. (If, for example, source databases and ontologies differentiated exact cross-references from non-exact cross references, Wikidata would be able to more precisely model these relationships.)

We have clarified these points in the manuscript.

Additionally, it would be useful to see a stricter comparison with sources like bridgedb, in terms of completeness and consistency.

We believe that a quantitative comparison to other mapping resources like BridgeDb would not be meaningful for a variety of reasons. First, both Wikidata and BridgeDb aggregate mappings from other 'authoritative' community resources (e.g., Ensembl for genes and proteins, ChEBI for chemicals, etc.). So a comparison between Wikidata and BridgeDb would really boil down to a comparison of the resources that were imported into each system. Second, for chemical compounds, BridgeDb actually imports mappings from Wikidata because it has a clear and transparent data model, and it allows people to fix inconsistencies and add missing content. Third, the scope of these resources is drastically different. BridgeDb currently focuses on genes, proteins, metabolic reactions, and metabolites, while Wikidata also includes many additional closely-related entity types (e.g., variants, diseases, organisms) as well as many more distantly-related types (e.g. clinical trials, people, countries).

7. Trustworthiness of curation: The strength of Wikidata is in the crowdsourcing of knowledge. While this can be scaled up more easily due less of a need for funded curators, the downside is that the information may be less accurate and reliable. The paper provides good evidence of reliability via the disease cross-referencing experiment, in which 99% of 2030 crowd-sourced mappings were reviewed and accepted.I am a little skeptical of the generalizability of these results. Mappings are generally quite easy and can be done with reasonable reliability by an automated string-matching process (with some caveats). It would be useful to know more about how many different people contributed to the 2030 (was it one person running a script)?Refinements to the ontology structure are harder, and it's not clear how often these were incorporated. The mapping results are still a nice example, but they need to be qualified more, and there should be more discussion on reliability.

We agree that identifier mapping represents the most common community contribution, and likely the most accessible crowdsourcing task. And the majority of the disease identifier mappings were performed by a handful of editors through the automated Mix'n'Match ID mapping interface (currently seven users with > 20 mappings). So while *disease* mapping through Mix'n'Match is skewed to a relatively small number of editors, the broader identifier mapping activities across all of Mix'n'Match is distributed across a much larger set of users, each with expertise and/or interest in a different area.

Regarding automated mapping via string matching, we agree that in many instances, automated methods are effective. However, in the case of the 2030 proposed MeSH and GARD mappings we reported, 771 (38%) were based on something other than simple string matching. We believe this supports the idea that crowdsourcing human mappings from Wikidata is a useful bridge between automated methods and expert curation.

We chose to focus on identifier mapping because the evaluation of accuracy by expert curators was relatively straightforward. We agree the refinements to ontology structure and the addition of other statements are likely to be more difficult and error-prone, and as we allude to in the manuscript, much more difficult to quantitatively evaluate for accuracy.

We have clarified all of these points in the manuscript.

8. Phenomizer analysis: It's not clear if there is circularity in the Phenomizer analysis:- were there publications that incorporated the case reports that were then annotated? Was this controlled for?- What happened on 2018-09 when the HPO-only semsim score jumps? Were the WD annotations incorporated into HPO? Or was this independent annotation? The subsequent drop is curious.

As part of a separate manuscript focused on NGLY1 deficiency (in revision), we curated disease-related phenotypes from two papers by Enns et al. (Enns et al., 2014) and Lam et al. (Lam et al., 2017, p. 1). These phenotypes were added as statements in Wikidata. In our analysis for this paper, we extracted phenotypes from suspected cases of CDDG / NGLY1-deficiency described in a third paper by Caglayan et al. (Caglayan et al., 2015) and used those as input to the Phenomizer analysis. So while our team was responsible for the majority of curation of phenotypes associated with NGLY1 deficiency, there was no "circularity" in the sense that different phenotype sets were used as the Phenomizer input and Wikidata additions. We have revised the manuscript to clarify this point.

After curation of the Enns et al. and Lam et al. papers, we also submitted those annotations for review and curation by curators with the Human Phenotype Ontology (https://github.com/obophenotype/human-phenotype-ontology/issues/3287). Based on annotations contributed by us and others, new phenotypes associated with CDDG / NGLY1-deficiency were added over the period shown in Figure 4. The sharp increase in the probability score around 2018-09 was primarily due to the addition of corneal ulceration (HP:0012804), which is a phenotype that is relatively specific for CDDG / NGLY1-deficiency and one of the phenotypes from Caglayan et al. that served as an input to the analysis. While there is a slight decline in the HPO-only probability score after 2018-09, we do not believe that reflects substantial changes to the underlying phenotypes annotated to CDDG / NGLY1-deficiency.

9. Some lines read as if written for a grant proposal, e.g., "Wikidata has a proven track record for leveraging...", "Wikimedia Foundation... has a long track record of developing and maintaining..." see Introduction. In many instances, Wikidata is presented as "unique". Please reword such sentences/passages.

We believe that the substance of those sentences is both correct and important to emphasize to readers for several reasons. First, many initiatives claim to be based on "crowdsourcing", but the vast majority fail to build and/or sustain a critical mass of users. It is noteworthy that the Wikimedia Foundation (WMF) has been successful at this on multiple occasions, and that Wikidata is one such example of a mature community-based resource.Second, the bioinformatics community continually generates innovative new tools, but the scalability of these tools are rarely tested with large numbers of users and large datasets. The WMF runs several online information resources (including Wikipedia and Wikidata) that run at a high level of performance and availability, which are important features for information management infrastructure. Third, Wikidata is sustained by funding streams that are different from the vast majority of biomedical resources (which are mostly funded by the NIH). Insulation from the 4-5 year funding cycles that are typical of NIH-funded biomedical resources does make Wikidata quite unique.

We have reworded several sections to hopefully strike a better balance between factual observation and advocacy. However, the intent of these sentences was not (and is not) to gratuitously praise Wikidata, but to convey to the reader that Wikidata is different than other online biomedical resources in several fundamental ways. We believe these points are important to communicate to readers as they decide how much effort to devote to using and/or contributing to Wikidata.

CODE/DATAThe Phenomizer analysis can't be reproduced at the moment, as the curated case reports with HPO IDs are not made available. Additionally, the settings for Phenomizer should be made available, also Phenomizer provides p-values, it would be useful to see this in the analysis results.

On more careful examination, we realized that we were imprecise with how we described our analysis. The published version of Phenomizer (Köhler et al., 2009) and the original Phenomizer web tool (http://compbio.charite.de/phenomizer/) are based on an "ontological similarity search", which includes the computation of a p-value. The most recent version of Phenomizer ("Orphamizer"; http://compbio.charite.de/phenomizer_orphanet/) is based on a newer algorithm for Bayesian network inference (Bauer et al., 2012) and a software package called BOQA (Köhler, 2020), which outputs probability scores. Our results are based on a BOQA analysis (with and without phenotype annotations from Wikidata). We have updated the manuscript text accordingly.

With that clarification, a Jupyter notebook that demonstrates the entire BOQA analysis from raw files to published figure has been added to our previously-cited Github repository: https://github.com/SuLab/Wikidata-phenomizer/ (Tu et al., 2019) (now also archived at (Tu et al., 2020)).

**References**

Bauer S, Köhler S, Schulz MH, Robinson PN. 2012. Bayesian ontology querying for accurate and noise-tolerant semantic searches. Bioinforma Oxf Engl 28:2502–2508. doi:10.1093/bioinformatics/bts471

Caglayan AO, Comu S, Baranoski JF, Parman Y, Kaymakçalan H, Akgumus GT, Caglar C, Dolen D, Erson-Omay EZ, Harmanci AS, Mishra-Gorur K, Freeze HH, Yasuno K, Bilguvar K, Gunel M. 2015. NGLY1 mutation causes neuromotor impairment, intellectual disability, and neuropathy. Eur J Med Genet 58:39–43. doi:10.1016/j.ejmg.2014.08.008

Enns GM, Shashi V, Bainbridge M, Gambello MJ, Zahir FR, Bast T, Crimian R, Schoch K, Platt J, Cox R, Bernstein JA, Scavina M, Walter RS, Bibb A, Jones M, Hegde M, Graham BH, Need AC, Oviedo A, Schaaf CP, Boyle S, Butte AJ, Chen Rui, Chen Rong, Clark MJ, Haraksingh R, FORGE Canada Consortium, Cowan TM, He P, Langlois S, Zoghbi HY, Snyder M, Gibbs RA, Freeze HH, Goldstein DB. 2014. Mutations in NGLY1 cause an inherited disorder of the endoplasmic reticulum-associated degradation pathway. Genet Med Off J Am Coll Med Genet 16:751–758. doi:10.1038/gim.2014.22

Köhler S. 2020. BOQA | Phenomics and Machine Learning @ Berlin. https://phenomics.github.io/software-boqa.html

Köhler S, Schulz MH, Krawitz P, Bauer S, Dölken S, Ott CE, Mundlos C, Horn D, Mundlos S, Robinson PN. 2009. Clinical diagnostics in human genetics with semantic similarity searches in ontologies. Am J Hum Genet 85:457–464. doi:10.1016/j.ajhg.2009.09.003

Lam C, Ferreira C, Krasnewich D, Toro C, Latham L, Zein WM, Lehky T, Brewer C, Baker EH, Thurm A, Farmer CA, Rosenzweig SD, Lyons JJ, Schreiber JM, Gropman A, Lingala S, Ghany MG, Solomon B, Macnamara E, Davids M, Stratakis CA, Kimonis V, Gahl WA, Wolfe L. 2017. Prospective phenotyping of NGLY1-CDDG, the first congenital disorder of deglycosylation. Genet Med Off J Am Coll Med Genet 19:160–168. doi:10.1038/gim.2016.75

Mungall CJ. 2019. Never mind the logix: taming the semantic anarchy of mappings in ontologies. Monkeying OWL. https://douroucouli.wordpress.com/2019/05/27/never-mind-the-logix-taming-the-semantic-anarchy-of-mappings-in-ontologie/

Tu R, Stupp GS, Su AI. 2020. SuLab/Wikidata-phenomizer: Release v1.0 on 2020-01-15. Zenodo. doi:10.5281/zenodo.3609142

Tu R, Stupp GS, Su AI. 2019. SuLab/Wikidata-phenomizer 7b25781. https://github.com/SuLab/Wikidata-phenomizer

Wikimedia Foundation. 2020. Wikimedia Statistics - Wikidata - Edits. https://stats.wikimedia.org/v2/#/wikidata.org/contributing/edits/normal|line|2-year|editor_type~anonymous*group-bot*name-bot*user|monthly